# VRSBench: A Versatile Vision-Language Benchmark Dataset for Remote Sensing Image Understanding

**Xiang Li**    **Jian Ding**    **Mohamed Elhoseiny**
King Abdullah University of Science and Technology
{xiang.li.1,jian.ding,mohamed.elhoseiny}@kaust.edu.sa

## Abstract

We introduce a new benchmark designed to advance the development of general-purpose, large-scale vision-language models for remote sensing images. Although several vision-language datasets in remote sensing have been proposed to pursue this goal, existing datasets are typically tailored to single tasks, lack detailed object information, or suffer from inadequate quality control. Exploring these improvement opportunities, we present a **V**ersatile vision-language **Bench**mark for **R**emote **S**ensing image understanding, termed **VRSBench**. This benchmark comprises 29,614  images, with 29,614 human-verified detailed captions, 52,472 object references, and 123,221 question-answer pairs. It facilitates the training and evaluation of vision-language models across a broad spectrum of remote sensing image understanding tasks. We further evaluated state-of-the-art models on this benchmark for three vision-language tasks: image captioning, visual grounding, and visual question answering. Our work aims to significantly contribute to the development of advanced vision-language models in the field of remote sensing. The data and code can be accessed at `https://vrsbench.github.io`.

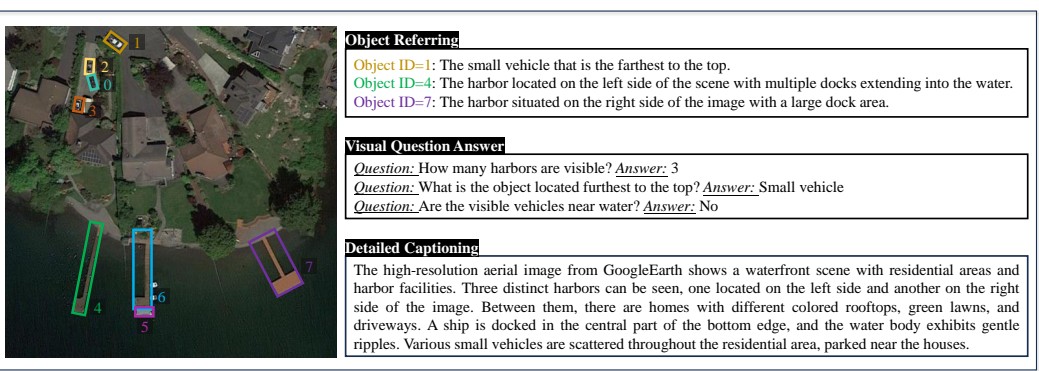

Figure 1: Examples of an image and corresponding annotations in VRSBench dataset. Our annotations include object referring, visual question answering, and detailed captions.

## 1   Introduction

Remote sensing models seek to understand the Earth's surface using imagery captured from overhead, offering a unique perspective of our physical world. This technique is instrumental in various applications, such as land use mapping, urban planning, precision agriculture, disaster management, etc. In the past few years, the success of large vision-language models (LVLMs)[1, 2, 3, 4, 5] in natural

38th Conference on Neural Information Processing Systems (NeurIPS 2024) Track on Datasets and Benchmarks.

scenes has inspired a trend of applying LVLMs to remote sensing [6]. Recent efforts have explored LVLMs for various remote sensing image understanding tasks, including scene classification [7, 8], image captioning [9, 10, 11, 12, 13, 14, 15, 16, 17], visual grounding [18, 19, 20], visual question answering (VQA) [21, 22, 23, 24, 25, 26], and general-purpose models [27, 28], etc.

However, directly applying LVLMs to remote sensing images presents challenges. LVLMs are typically trained on *internet data*, which differs significantly from remote sensing data. Remote sensing images often feature very small objects (sometimes only 10 pixels) and require complex spatial reasoning from an overhead view. Building effective LVLMs for remote sensing requires large-scale, high-quality datasets tailored to this field. Recent works[27, 28, 29] have attempted to train LVLMs with a combination of existing text-enriched remote sensing data, achieving reasonable performance. However, further improvements are limited by the current vision-language datasets in remote sensing, which have the following limitations:

**(i)** Existing vision-language datasets primarily cater to single image perception tasks, e.g., image captioning. Recent works explore integrating multiple datasets to accommodate a wider array of tasks [28, 29]. Such integration, while crucial, introduces challenges including inconsistent data annotations, variations in data quality, and the complexity of merging different data formats and sources, all of which can hinder model performance and scalability.

**(ii)** Most commonly used remote sensing image caption datasets, such as UCM-Captions [30] and RSICD [10], provide only brief descriptions, lacking detailed object information. Recent work RSGPT [27] provides high-quality, human-generated detailed image captions; however, the dataset comprises only 2,585 image-text pairs. This limited scope restricts its potential for training robust vision-language models in remote sensing applications. Although recent works, such as RS5M [31] and RemoteClip [7], introduced large-scale remote sensing image-text pair datasets, these annotations are automatically generated by image caption models and lack human verification. Given the current limitations of automatic captioning technology, such image-text data often suffer from accuracy issues and a lack of quality control.

**(iii)** Most existing remote sensing visual grounding datasets are designed under simplistic scenarios where the referring objects typically stand alone within their category. For instance, in the widely used DIOR-RSVG [19] datasets, a large portion of objects are unique within the categories, which leads to 38.36% of objects being easily distinguished by the object category alone. Finally, the majority of current VQA datasets in remote sensing employ automated methods for generating question-answer pairs. These automatically generated pairs often encompass a limited variety of unique questions and answers, which may not be sufficiently diverse to facilitate open-ended question-answering in real-world applications.

In this study, to address these limitations, we introduce a novel versatile benchmark for vision-language understanding of remote sensing images. VRSBench comprises 29,614 images, each enriched with human-verified detailed captions, complex object referring, and question-answer pairs, check Table 1 for a detailed comparison with existing datasets. This dataset facilitates the training and evaluation of vision-language models across a spectrum of remote sensing image understanding tasks. Fig. 1 gives an example of a selected image and associated annotations.

The key contributions of our work are summarized as follows:

- We introduce a new semi-automatic vision-language data collection pipeline which includes four key steps: object attributes extraction, prompt engineering, GPT-4 inference, and human verification. This pipeline enables a fast collection of large-scale datasets with human-level annotation quality.

- Based on the semi-automatic data collection pipeline, we collect VRSBench dataset that provides detailed image captioning, visual grounding, and visual question-answer labels in a unified dataset, and therefore, enables a comprehensive evaluation of multiple vision-languages capabilities based on this dataset.

- VRSBench provides large-scale human-verified annotations that feature several advantages: 1) it provides a large-scale collection of human-verified, high-quality captions rich in object details; 2) it offers more realistic object refers in which each referring sentence unambiguously identifies an object among multiple similar ones within the same category; 3) it features a diverse collection of open-ended question-answer pairs in natural language.

- We develop three benchmarks based on our VRSBench dataset, including detailed image caption, visual grounding, and visual question answering, and evaluate the performance of several state-of-the-art LVLMs.

Table 1: Comparison between existing remote sensing vision-language datasets and our VRSBench dataset. Values in parentheses in the Caption column indicate the average number of words in captions. OBB denotes orientated bounding box. A small portion of question-answer pairs in RSIVQA are annotated by human annotators.

| Dataset | Year | #Image | Caption | | Grounding | | VQA | | Human |
|---|---|---|---|---|---|---|---|---|---|
| | | | #Captions | Details | #Refers | OBB | #VQAs | Open-ended | |
| UCM-Captions [30] | 2016 | 2,100 | 10,500 (12) | ✗ | 0 | ✗ | 0 | - | ✓ |
| RSICD [10] | 2017 | 10,921 | 54605 (12) | ✗ | 0 | ✗ | 0 | - | ✓ |
| RS5M [31] | 2023 | 5M | 5M (49) | ✓ | 0 | ✗ | 0 | - | ✗ |
| RSICap [27] | 2023 | 2,585 | 2,585 (60) | ✓ | 0 | ✗ | 0 | - | ✓ |
| RSVG [18] | 2022 | 4,239 | 0 | ✗ | 7,933 | ✗ | 0 | - | ✓ |
| DIOR-RSVG [19] | 2023 | 17,402 | 0 | ✗ | 38,320 | ✗ | 0 | - | ✓ |
| RRSIS-D [20] | 2024 | 17,402 | 0 | ✗ | 17,402 | ✓ | 0 | - | ✓ |
| RSVQA-HR [21] | 2020 | 10,659 | 0 | ✗ | 0 | ✗ | 1,066,316 | ✗ | ✗ |
| RSIVQA [22] | 2021 | 37,264 | 0 | ✗ | 0 | ✗ | 111,134 | ✓ | ✓ |
| VQA-TextRS [24] | 2022 | 2,144 | 0 | ✗ | 0 | ✗ | 6,245 | ✓ | ✓ |
| RSIEval [27] | 2023 | 100 | 0 | ✗ | 0 | ✗ | 933 | ✓ | ✓ |
| VRSBench | 2024 | 29,614 | 29,614 (52) | ✓ | 52,472 | ✓ | 123,221 | ✓ | ✓ |

## 2 Pipeline

To construct our VRSBench dataset, we employed multiple data engineering steps, including attribute extraction, prompting engineering, GPT-4 inference [32], and human verification. These processes are meticulously designed to ensure the integrity and utility of the dataset for remote sensing applications.

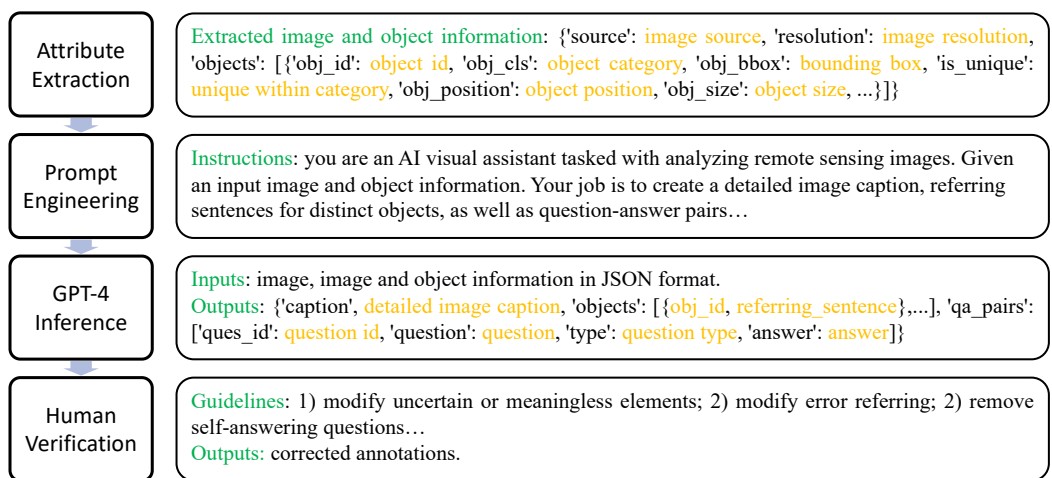

Figure 2: Dataset creation pipeline. We generate object information from detection labels and use carefully designed instructions to prompt GPT-4 to generate annotations from input images along with object information. All annotations are verified by human annotators.

### 2.1 Attribute Extraction

Initially, we extract image information, including the source and resolution, as well as object information—such as the object category, bounding box, color, position (absolute and relative), and size (absolute and relative)—from existing object detection datasets. We also determine whether an object is unique within its category, which is important for crafting accurate reference sentences.

In this study, we utilize two prominent open-access object detection datasets, DOTA-v2 [33] and DIOR [34], to develop our VRSBench dataset. Due to the unavailability of test labels for DOTA-v2, we incorporate only its training and validation sets. We divide each image into patches measuring 512 × 512 pixels. Notably, each image patch from DOTA-v2 contains, on average, 14.2 instances, while each patch from the DIOR dataset averages only 3.3 instances. This higher instance density in DOTA-v2 offers a more challenging and diverse training environment compared to existing remote sensing visual grounding datasets, such as DIOR-RSVG [19] and RRSIS-D [35], that are typically sourced from the DIOR dataset. Moreover, the visual grounding task predominantly involves identifying horizontal bounding boxes (HBB) based on referential descriptions. By constructing our dataset upon the framework of DOTA-v2, VRSBench facilitates the grounding of objects with orientated bounding boxes (OBB), thereby extending the capabilities of traditional visual grounding methods.

## 2.2 Prompt Engineering

We carefully design the following instructions to prompt GPT-4V to create detailed image captions, object referring, and question-answer pairs. Detailed instructions for each task are provided in the supplementary.

"You are an AI visual assistant tasked with analyzing remote sensing images. For each image, you receive image meta information and a list of objects in the format of .... Your job is to create a detailed image caption and referring sentences for 1-5 distinct objects, if multiple are present, as well as 3-10 question-answer pairs. Each referring sentence should unambiguously refer to one object. Finally, you need to return a JSON file in the format: {caption: detailed image caption, objects: [obj_id: object id, ref: referring sentence,...], qa_pairs: [ques_id: question id, question: question, type: question type, answer: answer]}. Do not return any notes after the JSON."

## 2.3 GPT-4V Inference

Given input prompts, we call OpenAI API[1] to automatically generate annotations. We iteratively refine our instructional prompts to generate annotations, meticulously enhancing these instructions to ensure the quality of the annotations. In the responses generated by GPT-4V, undesirable terms, such as "not provide", "not specified", and "unknown", may be present. Should any of the specified excluding phrases appear in GPT-4V's output, the procedure requires that GPT-4V be recursively invoked to regenerate responses until the output is free of any excluding phrases. This iterative process is attempted a maximum of five times, after which the final response is utilized for generating annotations. Ultimately, any caption sentences, object-referring sentences, or question-answer pairs containing these excluding phrases are excised from final annotations.

## 2.4 Human Verification

With our carefully designed prompts, most of the annotations generated by GPT-4V are accurate. Nevertheless, a significant number of outputs remain suboptimal. This shortfall is likely attributable to the model's limited exposure to remote sensing imagery, which impedes its capacity to interpret complex structures within these images. Additionally, it is important to note that even advanced language models, such as the GPT-4V system, exhibit a degree of hallucinatory outputs [36].

To improve the quality of the dataset, we engage human annotators to validate each annotation generated by GPT-4V. This validation process incorporates domain experts to guarantee that annotators have a comprehensive understanding of the assigned tasks. Initially, domain experts establish detailed guidelines, which include directives such as: 1) eliminate any uncertain or irrelevant elements; 2) ensure each referring sentence unambiguously identifies the intended object; 3) exclude questions that inherently contain their answers. More details about the annotation guidelines can be found in the supplementary. The verification of each image requires approximately 120 seconds, culminating in a total of 1,004 hours devoted to human verification. Each image verification costs around 0.21 USD and leads to a total cost of 6,200 USD for human verification. To enhance the quality of our dataset, we have instituted a secondary validation phase involving a meticulous re-evaluation of 2,000 images. This step is designed to uncover prevalent annotation discrepancies and to refine the annotators' understanding of the task requirements.

---

[1]https://platform.openai.com/docs/api-reference

# 3 VRSBench Dataset

## 3.1 Dataset Overview

Our VRSBench dataset contains 29,614 remote sensing images, with high-quality human-verified annotations. It comprises 29,614 caption sentences, 52,472 referring sentences, and 123,221 question-answer pairs. Each image is of $512 \times 512$ pixels. Details of each type of annotation are given below. Note that original object detection labels and object attributes are also provided in our annotations.

## 3.2 Detailed Caption

VRSBench captions provide comprehensive descriptions that encompass both abstract image attributes and detailed object-specific information. Each caption initiates with a general overview of the image, subsequently delving into explicit and precise details present within the image. Attributes of the image include the source, resolution, color or panchromatic distinction, and the type of scene depicted. Conversely, object attributes cover object quantity, color, shape, size, and spatial positioning of each object, encompassing both its absolute location within the image and its relative positioning in relation to other objects. Descriptions are confined to manifest features, eschewing any elements that are uncertain or ambiguous. Additionally, captions may incorporate other visually discernible objects not supplied by the source object detection datasets, such as buildings, houses, roads, and trees, if these elements are clear and unambiguous. Each caption typically comprises 3-7 sentences, with an average length of 54 words. A summary of these caption statistics is detailed in Fig. 3.

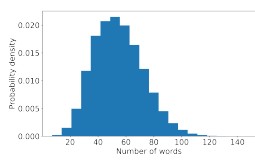

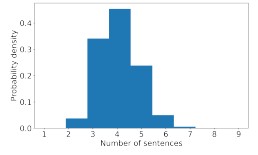

| #images | 29,614 |
|---|---|
| #vocabulary size | 9,588 |
| #total words | 1,526,338 |
| #caption sentences | 114,366 |
| Avg. #sentences in caption | 4 |
| Avg. caption length | 52 |

|(a) PDF of word count|(b) PDF of sentence number|(c) Statistics of VRSBench captions.|

Figure 3: Statistics of the VRSBench caption dataset. (a) Probability density function (PDF) of caption length. (b) PDF of the sentence number. (c) Summative statistics.

## 3.3 Object Referring

In VRSBench, each image is analyzed to identify 1-5 distinct objects, and referring sentences are provided for each. These sentences are carefully crafted such that each can independently and unambiguously identify an object without reliance on other sentences. We utilize distinctive features to clearly differentiate the referred objects from others within the image. These features span a variety of object attributes including color, shape, position, size, relative position, and relative size, among others. Note that the original DOTA-v2 and DIOR datasets contain 18 and 20 object categories respectively, which are merged into 26 object categories in our dataset. Please check the supplementary for category merging details. Figure 4 provides a summary of referring sentences of our VRSBench dataset.

## 3.4 Visual Question Answering

Based on all visible elements and object information, we provide 3-10 question-answer pairs about diverse types, including object category, object existence, object quantity, object color, object size, object position, direction, scene characteristics, and complex reasoning, and provide an answer for each question. Instead of only focusing on objects from source detection datasets, we also ask questions about objects that are not provided, such as houses, roads, and trees if they are obvious and non-ambiguous. When collecting annotations, we ensure each question has a definite answer without any ambiguity, and answer each question using a single word or phrase. We show the statistics of question-answer pairs in Figure 5.

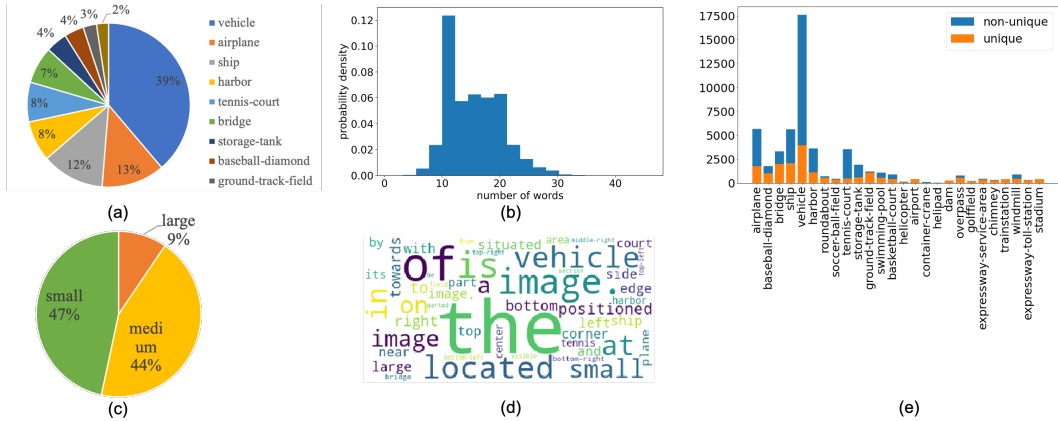

Figure 4: Statistics of object referring sentences of VRSBench dataset. (a) Distribution of the 10 most frequent object categories. (b) Distribution of the word length of referring sentences. (c) Distribution of object size. (d)Word cloud of the top 50 words in referring sentences. (e) Distribution of unique/non-unique objects in each category.

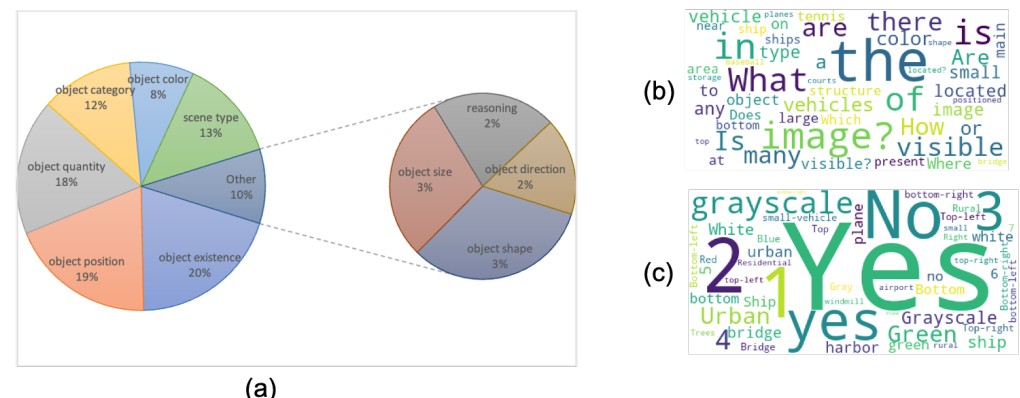

Figure 5: Statistics of question-answer pairs in VRSBench. (a) Distribution of question types. (b) Word cloud of top 50 most frequent words in questions. (c) Word cloud of top 50 most frequent words in answers.

## 4 Benchmark Evaluation

### 4.1 Benchmark Overview

Based on VRSBench, we construct three distinct tasks for advancing remote sensing image understanding:

- VRSBench-Cap: This challenge requires the prediction of a comprehensive description for a given remote sensing image, encapsulating intricate object details and contextual relevance.
- VRSBench-Ref: The task involves identifying and localizing specific objects from a given remote sensing image based on textual descriptions.
- VRSBench-VQA: This task aims to answer questions related to visual content in a given remote sensing image.

To facilitate benchmark evaluation, we partition our VRSBench dataset into two distinct, non-overlapping splits designated for model training and evaluation. We split the datasets according to official splits of DOTA [33] and DIOR [34] datasets, where their training images are used to build the training set of VRSBench and their validation sets are used as the test set. Table 2 delineates the statistics of two splits.

For the above three tasks, we benchmark state-of-the-art models, including LLaVA-1.5 [37], MiniGPT-v2 [38], and GeoChat [28], to demonstrate the potential of LVLMs for remote sensing image understanding. LLaVA-1.5 [37], MiniGPT-v2 [38], Mini-Gemini [2], and GeoChat [28] are generalist models that are naturally designed for general-proposed image understanding. We, therefore, report the performance of these methods under joint training of all three tasks, i.e., image captioning, visual grounding, and VQA.

Table 2: VRSBench data split.

|          | train  | test   |
|----------|--------|--------|
| #Images   | 20,264 | 9,350  |
| #Captions | 20,264 | 9,350  |
| #Refers   | 36,313 | 16,159 |
| #VQAs     | 85,813 | 37,408 |

Note that all these comparison methods include a multi-stage training process. To ensure a fair comparison, we reload the models that are initially trained on large-scale image-text alignment datasets, and then finetune each method using the training set of our VRSBench dataset. We employ CLIP-ViT(L-14) [39] as the vision encoder and use the Vicuna-7B model [40] as the Large Language Model (LLM). For LLaVA-1.5 [37], Mini-Gemini [2], and GeoChat [28], we adhere to the original model specifications, utilizing two MLP layers with GeLU activation [41]. For MiniGPT-v2[38], we implement a single MLP layer as described in the original paper. For each comparing method, we finetune the model on the training set of VRSBench dataset for 5 epochs. Following GeoChat [28], we use LoRA [42] finetuning to finetune all comparing methods, with a rank of 64. To understand the benefit of fientuning on VRSBench, we include the baseline GeoChat [28] without training on our VRSBench dataset for comparison.

We further evaluate the performance of GPT-4V, which is generally known as one of the most powerful close-source vision-language models, on three tasks based on our VRSBench dataset. To achieve this, we directly call GPT-4V API to generate detailed captions, referring object locations, and answers for visual questions, with the following instructions. Note that we do not include object information in this experiment.

## 4.2 Detailed Image Caption

**Evaluation metrics**. For model evaluation, we follow standard practices by utilizing a set of established metrics including BLEU [43], ROUGE_L [44], METEOR [45], and CIDEr [46]. For BLEU, we consider n-gram precision with n values of 1, 2, 3, and 4. We also report average caption lengths to assess the details of generated captions. Furthermore, we note that traditional caption evaluation metrics may not be suitable for long captions. We, therefore, use an LLM-based caption evaluation metric called CLAIR[2] proposed in [47] for our detailed image caption task.

**Results.** Table 3 shows the comparative performance of different methods in detailed image captioning of our VRSBench dataset. As demonstrated in the table, the baseline GeoChat model, when not finetuned on the VRSBench dataset, exhibits significantly poorer performance compared to models that have been finetuned on VRSBench. The LLaVA-1.5 [37] model that has undergone fine-tuning on VRSBench achieves the highest performance, reaching a BLEU-1 score of 48.1 and a CIDEr score of 33.9. The GPT-4V model shows the best performance on the CLAIR score. This is expected because the CHIAR score itself is calculated using GPT-4 in our experiments. Moreover, the generated captions have an average word length of 49, which closely approximates the average length of ground truth captions. Note that detailed image captioning is a more challenging task than conventional image caption, therefore, the performance falls far behind. More advanced vision-language modeling techniques are desired to handle this challenging task.

Table 3: Detailed image caption performance on VRSBench dataset. Avg_L denotes the average word length of generated captions. Boldface indicates the best performance.

| Method | BLEU-1 | BLEU-2 | BLEU-3 | BLEU-4 | METEOR | ROUGE_L | CIDEr | CLAIR | Avg_L |
|--------|--------|--------|--------|--------|--------|---------|-------|-------|-------|
| GeoChat w/o ft [28] | 13.9 | 6.6 | 3.0 | 1.4 | 7.8 | 13.2 | 0.4 | 0.42 | 36 |
| GPT-4V [32] | 37.2 | 22.5 | 13.7 | 8.6 | 20.9 | 30.1 | 19.1 | **0.83** | 67 |
| MiniGPT-v2 [38] | 36.8 | 22.4 | 13.9 | 8.7 | 17.1 | 30.8 | 21.4 | 0.73 | 37 |
| LLaVA-1.5 [37] | **48.1** | **31.5** | **21.2** | **14.7** | **21.9** | **36.9** | **33.9** | 0.78 | 49 |
| GeoChat [28] | 46.7 | 30.2 | 20.1 | 13.8 | 21.1 | 35.2 | 28.2 | 0.77 | 52 |
| Mini-Gemini [48] | 47.6 | 31.1 | 20.9 | 14.3 | 21.5 | 36.8 | 33.5 | 0.77 | 47 |

---

[2]https://github.com/davidmchan/clair

## 4.3 Visual Grounding

**Benchmark settings**. In this study, we focus on the grounded localization task, which aims to predict bounding boxes for referring objects. In our experiments, we use horizontal bounding boxes for model training and evaluating the grounding performance. Results on OBB visual grounding can be found in the supplementary.

**Evaluation metrics**. For model evaluation, we employ the metric accuracy@$\tau$ to assess performance. Accuracy is determined by calculating the Intersection over Union (IoU) between the predicted bounding box and the ground-truth box. A prediction is considered accurate if the IoU exceeds the threshold $\tau$. In our experiments, we choose two different IoU thresholds, i.e., 0.5 and 0.7.

**Results.** Table 3 shows the visual grounding performance of different methods on our VRSBench dataset. From the table, the model finetuned on the VRSBench significantly outperforms the baseline GeoChat model without finetuning. Furthermore, all models demonstrate superior performance in tasks involving unique object referring compared to non-unique object referring. This superiority is expected, as it is generally easier to localize objects uniquely identified within their categories than to differentiate among multiple instances within the same category. Note that even though MiniGPT-v2 gets worse overall grounding performance, it performs better at grounding non-unique objects.

Furthermore, GPT-4V exhibits markedly inferior performance compared to models specifically trained on image captioning and visual grounding tasks, primarily due to the absence of object information in its prompts. Despite the notable successes of existing closed-source multimodal large language models (LLMs), such as GPT-4, in comprehending natural images, their effectiveness is notably reduced when not fine-tuned on remote sensing imagery.

More importantly, even the best-performing GeoChat model fails to achieve satisfactory performance levels, with a grounding accuracy of 49.8% at a threshold of 0.5. This shortfall is attributed to the demanding scenarios presented in the VRSBench dataset, which includes multiple instances of the same category as the target object. This highlights the necessity for more advanced vision grounding techniques to effectively tackle these complexities.

Table 4: Visual grounding performance on VRSBench dataset. Boldface indicates the best performance.

| Method | Unique | | Non Unique | | All | |
|---|---|---|---|---|---|---|
| | Acc@0.5 | Acc@0.7 | Acc@0.5 | Acc@0.7 | Acc@0.5 | Acc@0.7 |
| GeoChat w/o ft [28] | 20.7 | 5.4 | 7.3 | 1.7 | 12.9 | 3.2 |
| GPT-4V [32] | 8.6 | 2.2 | 2.5 | 0.4 | 5.1 | 1.1 |
| MiniGPT-v2 [38] | 40.7 | 18.9 | 32.4 | 15.2 | 35.8 | 16.8 |
| LLaVA-1.5 [37] | 51.1 | 16.4 | 34.8 | 11.5 | 41.6 | 13.6 |
| GeoChat [28] | **57.4** | **22.6** | **44.5** | **18.0** | **49.8** | **19.9** |
| Mini-Gemini [48] | 41.1 | 9.6 | 22.3 | 4.9 | 30.1 | 6.8 |

## 4.4 Visual question answering

**Evaluation metrics**. We categorize the questions in the test set into 10 distinct types: object category, presence, quantity, color, shape, size, position, direction, scene characteristic, and reasoning. The first eight categories relate to object-level questions, whereas the last two are aligned with scene-level, and reasoning-level questions, respectively. We present the overall accuracy as well as the accuracy for each individual question type. To ensure a robust evaluation, we use GPT-4 to determine for each question whether the answers match ground truth texts, with the prompt: *"Question: {question}, Ground Truth Answer: {ground_truth}, Predicted Answer: {predicted answer}. Does the predicted answer match the ground truth? Answer 1 for match and 0 for not match. Use semantic meaning not exact match. Synonyms are also treated as a match, e.g., pond and swimming pool."*.

**Results.** Table 5 shows the VQA performance of different methods on our VRSBench dataset. As shown in the table, the baseline GeoChat [28] model without finetuning gets an average accuracy of 40.8%. Further finetuning on our VRSBench training set significantly boosts the average accuracy to 76.0%. GPT-4V gets a reasonable performance but still falls a lot behind fine-tuned models,

suggesting that detailed object information contributes a lot to the visual question answering task on our benchmark.

Table 5: Visual question answering performance on VRSBench dataset. Boldface indicates the best performance.

| Method | Category | Presence | Quantity | Color | Shape | Size | Position | Direction | Scene | Reasoning | All |
|---|---|---|---|---|---|---|---|---|---|---|---|
| # VQAs | 5435 | 7789 | 6374 | 3550 | 1422 | 1011 | 5829 | 477 | 4620 | 902 | |
| GeoChat w/o ft [28] | 48.5 | 85.9 | 19.2 | 17.0 | 18.3 | 32.0 | 43.4 | 42.1 | 44.2 | 57.4 | 40.8 |
| GPT-4V [32] | 67.0 | 87.6 | 45.6 | 71.0 | 70.8 | 54.3 | 67.2 | 50.7 | 69.8 | 72.4 | 65.6 |
| MiniGPT-v2 [38] | 61.3 | 26.0 | 46.1 | 51.0 | 41.8 | 11.2 | 17.1 | 12.4 | 49.3 | 21.9 | 38.2 |
| LLaVA-1.5 [37] | 86.9 | 91.8 | 58.2 | 69.9 | 72.2 | **61.5** | **69.5** | **56.7** | **83.9** | 73.4 | 76.4 |
| GeoChat [28] | 86.5 | **92.1** | 56.3 | 70.1 | 73.8 | 60.4 | 69.3 | 53.5 | 83.7 | 73.5 | 76.0 |
| Mini-Gemini [48] | **87.8** | 92.1 | **58.8** | **74.0** | **75.3** | 58.0 | 68.0 | 56.7 | 83.2 | **74.4** | **77.8** |

# 5 Related Work

## 5.1 Remote Sensing Image Captioning Datasets

Image captioning in remote sensing is a well-established task that focuses on creating descriptive text for overhead imagery. Commonly used datasets such as UCM-Captions [30], Syndey-Captions [30], and RSICD [10] have been instrumental by offering brief scene descriptions. However, these datasets typically provide short and less detailed captions that overlook intricate object details. Recent efforts, such as RSGPT [27], have introduced high-quality, human-generated detailed captions, though the dataset is limited to just 2,585 image-text pairs, which hampers its utility for developing robust vision-language models in remote sensing. In contrast, RS5M [31] introduced a substantial dataset featuring 5 million detailed captions. However, these captions are generated automatically, resulting in quality that is not guaranteed. In stark contrast, our VRSBench dataset includes 29,614 human-verified captions that are not only of high quality but also rich in detail, ensuring both reliability and practical utility for advanced remote sensing applications.

## 5.2 Remote Sensing Visual Grounding Datasets

Visual grounding in remote sensing has recently emerged as an intriguing field of study. Unlike referring expressions in natural images, those in RSVG frequently involve complex geospatial relationships, and the objects of interest may not be prominently visible. The first RSVG dataset was introduced in [18], featuring 4,239 images from GoogleEarth and 7,993 referring expressions. Subsequently, Zhan et al. [19] introduced the DIOR-RSVG dataset, which includes 17,402 remote sensing images and 38,320 referring expressions across 20 object categories. Recent studies [35, 20] have developed visual grounding datasets for remote sensing that include object segmentation; however, these tend to be smaller in scale. In contrast, our VRSBench dataset incorporates a substantial number of object-referring expressions.

## 5.3 Remote Sensing Visual Question Answering Datasets

RSVQA [21] established the first VQA benchmark dataset for remote sensing images. This dataset comprises RS images sourced from OpenStreetMap, accompanied by automatically generated questions and answers. It includes 772 images with 77,232 question-answer pairs in the low-resolution collection and 10,659 images with 1,066,316 pairs in the high-resolution collection. Zheng et al.[22] launched the RSIVQA dataset, a remote sensing VQA dataset that features approximately 37k images and 110,000 question-answer pairs. A small portion of question-answer pairs in RSIVQA are annotated by human annotators. Al et al.[24] introduced an innovative dataset, VQA-TextRS, which consists of 2,144 RS images and 6,245 question-answer pairs generated and annotated by humans in an open-ended format. More recently, the RSIEval[27] dataset features 936 human-crafted question-answer pairs from 100 remote sensing images. Similarly, our VRSBench dataset also incorporates open-ended question-answer pairs, created by GPT-4V and validated by human annotators, with 123,221 question-answer pairs.

# 6 Conclusion and future work

In this work, we introduce VRSBench, a versatile vision-language dataset and benchmark for remote sensing image understanding. This comprehensive dataset not only addresses the limitations of previous datasets that either ignore detailed object information or suffer from quality control issues but also enriches the field by providing a diverse range of annotations including detailed captions, object referring, and visual question answering with rich object information and verified by human annotators. Our benchmark challenges, specifically designed around the VRSBench dataset, demonstrate the practical utility of our dataset in advancing the capabilities of vision-language models in the domain of remote sensing.

Currently, the VRSBench dataset is limited to annotations for RGB images. In future work, we aim to enhance VRSBench by incorporating annotations from a variety of remote sensing data types, including infrared images, multi- and hyperspectral images, Synthetic Aperture Radar (SAR) images, and temporal datasets. This expansion will significantly broaden the dataset's utility across diverse observation conditions, facilitating more accurate and timely applications in remote sensing.

# 7 Broader Impact

VRSBench provides a comprehensive benchmark for developing and evaluating generalist vision-language models in both remote sensing and computer vision. This dataset not only supports the training and evaluation of advanced vision-language models but also boosts their ability to tackle complex real-world scenarios in remote sensing.

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
