# Supplementary of VRSBench: A Versatile Benchmark for Vision Language Understanding of Remote Sensing Images

**Xiang Li    Jian Ding    Mohamed Elhoseiny**
King Abdullah University of Science and Technology
{xiang.li.1,jian.ding,mohamed.elhoseiny}@kaust.edu.sa

## 1 VRSBench Documentation and Intended Uses

### 1.1 Overview

VRSBench consists of 29,614 remote sensing images with detailed captions, 52,472 object refers, 123,221 visual question-answer pairs. VRSBench is designed to facilitate the development and evaluation of vision-language models in remote sensing, providing a comprehensive set of annotations including detailed captions, visual grounding, and visual question answering. This section documents the dataset in accordance with best practices to ensure transparency, reproducibility, and ethical usage.

### 1.2 Data Organizing

Our VRSBench dataset is organized as follows.

```
root/
├── Images_train.zip
├── Annotation_train.zip
├── Images_val.zip
├── Annotation_val.zip
├── VRSBench_train.json
├── VRSBench_EVAL_Cap.json
├── VRSBench_EVAL_referring.json
└── VRSBench_EVAL_vqa.json
```

Detailed descriptions for each folder or file are given below.

- Images_train.zip contains all raw images in the training split.

- Annotation_train.zip contains all annotations in the training split, one JSON file per image.

- Images_val.zip contains all raw images in the validation split.

- Annotation_val.zip contains all annotations in the validation split, one JSON file per image.

- VRSBench_train.json contains all training annotations following LLaVA in standard JSON format.

- VRSBench_EVAL_Cap.json contains all evaluation annotations for the captioning task in standard JSON format.

- VRSBench_EVAL_referring.json contains all evaluation annotations for the visual grounding task in standard JSON format.

- VRSBench_EVAL_vqa.json contains all evaluation annotations for the VQA task in standard JSON format.

38th Conference on Neural Information Processing Systems (NeurIPS 2024) Track on Datasets and Benchmarks.

## 1.3 Intended Uses

VRSBench is intended for use in academic and research settings, specifically for:

- Training and evaluating vision-language models capable of understanding complex visual and textual tasks.
- Advancing the state-of-the-art in remote sensing image analysis by providing a rich dataset that supports multiple tasks.

## 1.4 Use Cases

- **Academic Research**: VRSBench is ideal for exploring new algorithms in image captioning, visual grounding, and visual question answering within the remote sensing domain.
- **Model Evaluation**: The dataset can serve as a benchmark for comparing different vision-language models' performance on a standardized set of tasks.
- **Educational Purposes**: The dataset and its comprehensive annotations can be used in coursework and workshops to teach advanced techniques in machine learning and remote sensing.

## 1.5 Limitations

- **Geographic Diversity**: While VRSBench includes a variety of landscapes, the geographic diversity is limited to the regions covered by the DOTA-v2 and DIOR datasets.
- **Annotation Bias**: Despite efforts to ensure high-quality annotations through human verification, biases may exist in the interpretations of visual data due to subjective human factors.

## 1.6 Ethical Considerations

- **Privacy and Sensitivity**: The dataset consists of non-sensitive, publicly available satellite images where no individual person or private property can be identified.
- **Use Restrictions**: Users are encouraged to use VRSBench responsibly and ethically, particularly when developing applications that might impact environmental monitoring and urban planning.

## 1.7 Documentation and Maintenance

- **Versioning**: Detailed version history of the dataset will be maintained to track changes and improvements over time.
- **Community Involvement**: Feedback from the user community is encouraged to improve the dataset's quality and applicability in various use cases.

## 1.8 Statements for NLP

We employ GPT-4V [1] to generate initial annotations; for further details, please refer to the main paper. These annotations undergo a manual review by human annotators.

## 1.9 Accountability Framework

To ensure responsible usage and continuous improvement, an accountability framework is established. Users of VRSBench are encouraged to report any issues or biases they encounter, contributing to an ongoing effort to refine the dataset and its annotations.

# 2 Dataset Collection Details

- **Source datasets**: Images are sourced from the DOTA-v2 [2] and DIOR [3] datasets and annotated with high-resolution details. We divide each image into patches measuring 512 × 512 pixels and filter out patches with no object annotations. This yields over 20,310 image

patches from the DOTA-v2 dataset and 9,304 patches from the DIOR dataset. Statistics are given in Table 1.

- **Preprocessing**: We extract image-level information and object-level information for all image patches. Note that the original DOTA-v2 and DIOR datasets contain 18 and 20 object categories respectively. We merge shared object categories and also merge small-vehicle and large-vehicle into the vehicle category. After merging, we get 26 object categories, including airplane, airport, baseball-diamond, basketball-court, bridge, chimney, container-crane, dam, expressway-service-area, expressway-toll-station, golf-field, ground-track-field, harbor, helicopter, helipad, overpass, roundabout, ship, soccer-ball-field, stadium, storage-tank, swimming-pool, tennis-court, train-station, vehicle, windmill.

- **Object attribute extraction**: We then extract object attributes and formulate a JSON file for each image patch, including object category, corner points, bounding box, position, relative position, size, and relative size. We also determine whether each object is unique within its category or not. We do not extract object colors because objects can have complex structures with multiple colors, and we rely on GPT-4V to identify object colors. The code for preprocessing is provided at `https://github.com/lx709/VRSBench`.

- **GPT-4V annotation generation**: The code for GPT-4V inference is provided at `https://github.com/lx709/VRSBench`. Detailed instructions are provided in Section 6.1.

Table 1: Statistics of source object detection datasets.

| Dataset | #Images | #Valid Patches | #Selected Patches | Category |
|---|---|---|---|---|
| DOTA-v2 [2] | 2,423 | 29,910 | 20,310 | 18 |
| DIOR [3] | 11,725 | 9,304 | 9,304 | 20 |

## 3   URL to Data and Metadata

The VRSBench dataset can be accessed and downloaded through our dedicated platform, which provides detailed views of the dataset components and their annotations.

For practical examples and to download the dataset, visit our Huggingface repository (`https://huggingface.co/datasets/xiang709/VRSBench`). Detailed metadata for the dataset is documented using the Croissant metadata framework, ensuring comprehensive coverage and compliance with the MLCommons Croissant standards, check [metadata](`https://huggingface.co/api/datasets/xiang709/VRSBench`). Please check our Huggingface repo for metadata details.

## 4   Author Statement and Data License

**Author Responsibility Statement:** The authors bear all responsibilities in case of any violations of rights or ethical concerns regarding the VRSBench dataset.

**Data License Confirmation:** The dataset is released under the [CC-BY-4.0], which permits unrestricted use, distribution, and reproduction in any medium, provided the original work is properly cited.

## 5   Hosting and Accessibility

The VRSBench dataset is hosted on Huggingface (`https://huggingface.co/datasets/xiang709/VRSBench`) to ensure reliable and continuous accessibility.

**Maintenance Plan:** Ongoing maintenance and updates will be managed by the dataset authors, with updates scheduled bi-annually or as significant changes in the data sources occur.

**Long-term Preservation:** The dataset is archived in Huggingface (`https://huggingface.co/datasets/xiang709/VRSBench`) to ensure long-term availability.

**Structured Metadata:** The annotation for each image is well-organized in standard JSON format to ensure easy usage.

# 6 Data Creation Details

## 6.1 GPT-4V Prompts

We carefully design the following instructions to prompt GPT-4V to generate annotations of image captions, referring sentences, and visual question-answering pairs.

"You are an AI visual assistant tasked with analyzing remote sensing images. For each image, you receive image meta information and a list of objects in the format: {image source: image source, image resolution: image resolution, objects: [obj_id: object id, obj_cls: object category, obj_corner: corner point, obj_coord: object bounding box, is_unique: unique object or not, obj_position: object position, obj_rel_position: object relative position within category, obj_size: object size, obj_rel_size: object relative size within category, flag: refer or not, ...]}. The bounding box coordinates [x1, y1, x2, y2] are floating numbers from 0 to 1, corresponding to the top left x, top left y, bottom right x, and bottom right y. Note that the top-left corner coordinates are (0,0) and the bottom-right corner coordinates are (1,1).

Your job is to create a detailed image caption and referring sentences for 1-5 distinct objects, if multiple are present, as well as a list of question-answer pairs. Each referring sentence should unambiguously refer to one object. Finally, you need to return a JSON file in the format: {caption: detailed image caption, objects: [obj_id: object id, ref: referring sentence,...], qa_pairs: [ques_id: question id, question: question, type: question type, answer: answer]}. Do not return any notes after the JSON.

Here are further important instructions for referring sentences:
1. Identify 1-5 distinguishable objects and provide referring sentences. Each sentence alone must independently, without seeing others, and unambiguously identify an object.
2. Select all unique objects (is_unique=True) for creating referring sentences. Do not select objects whose flag=True for referring sentences, but still use them for captioning and question-answering tasks.
3. Use distinctive features to describe objects. Try to use diverse object attributes such as color, shape, position, size, relative position, and relative size, but avoid specifying size details for small or large vehicles. Some object attributes are not provided, you may need to identify them from the input image. Do not explain why it is distinctive or distinguishable.
4. For each object category, select only 1-3 most distinguishable objects and ensure the referring sentences can confidently distinguish each of them from other objects of the same category.
5. Avoid ordinal descriptors and references (first-mentioned, aforementioned, or previously mentioned) to prior mentions. Instead, use distinct features to refer back to previously identified objects.
6. If multiple object categories exist, try to include diverse object categories in a balanced manner.
7. For referring sentences, use natural language to describe objects based on their bounding box data, without directly mentioning the coordinates. Do not mention whether the object is distinguishable or not.
8. You may include roads/bridges running east-west or north-south but do not mention object-facing directions or pointing directions.
9. Do not mention the noses, vertical stabilizers, tails, or tail fins of planes, airplanes, or aircraft
10. Do not mention gate numbers when describing airports or airplanes.
11. Carefully verify each piece of information before finalizing the referring sentences, make sure each referring sentence alone can distinguish one object without any ambiguity. If not, remove this referring object.

Here are further important instructions for image captioning:
1. Create a detailed caption for the provided image, incorporating all visible elements and object information. Focus on describing the content of the image without mentioning the reference status of objects or their flag status.
2. Start the caption with an overview of the image. Possibly include the source of the image (if provided), specify whether it is in grayscale or color, and mention the resolution (if provided). Follow this with a description of specific, clear details within the image. Summarize the image's content in 3-7 sentences, making sure to include counts of prominent objects.
3. Describe only clear features; avoid uncertainties and unclear elements. Do not mention anything that is unknown or not specified.
4. Possibly include other visual objects in the image that are not provided as inputs, such as buildings,

houses, roads, and trees if they are obvious and non-ambiguous.

5. Highlight diverse object attributes such as color, shape, position, size, relative position, and relative size. Do not add size details for small or large vehicles.

6. Exclude imagined details not visible in the image, like weather, people, or object usage. Do not imagine the moving status of airplanes, ships, or vehicles if you are not sure about it.

7. For roads, include features like shape (straight/curved), width, length, and orientation.

8. For houses, mention characteristics like density, size, rooftop color, and presence of gardens.

9. For airports, include details like boarding bridges, terminals, boarding ports, and tarmac.

10. Carefully verify each piece of information before finalizing the caption.

11. Do not mention whether the image is taken during the day or night.

12. Do not mention whether the vehicles are in motion or not.

Here are questions for visual question answering:

1. Based on all visible elements and object information, ask 3-10 questions about diverse types, including object category, object existence, object quantity, object color, object shape, object size, object position, object direction, scene type, rural or urban, and reasoning. The category of scene type includes the main structure/type of area. Additionally, the category of reasoning is available for questions that require multifaceted analytical thought processes (e.g., object distribution pattern). Possibly include objects that are not provided, such as houses, roads, and trees if they are obvious and non-ambiguous.

2. Do not mention the object referred or not. Do not mention any flag information in questions and answers.

3. Ensure each question has a definite answer without any ambiguity, and answer each question using a single word or phrase, no more than 3 words.

4. When answering questions about the number of objects, take into account all object information.

5. Only ask questions about clear answers; avoid uncertainties or unclear elements, such as unknown, uncertain, some, or several. If the answer is uncertain or unable to be determined, remove this question-answer pair.

6. Do not use first, second, third, fourth, fifth, first-mentioned, or previously mentioned to refer to objects, use distinguishable features to refer to specific objects mentioned before.

7. Try to cover diverse types of questions.

8. Do not ask about the type of view the image was captured from, or whether the image was taken during day or night.

9. Do not ask the source of the image.

10. Do not ask facing direction, but you may ask whether roads/bridges running east-west or north-south.

11. Do not ask whether the vehicles are in motion or not.

12. Do not ask whether the image is taken during the day or night".

## 6.2 Human verification guidelines

Given an input image and associating detailed image caption, check if each piece of provided information is correct or not (no check for image source). If incorrect, correct the information, possible corrections include modifying/removing words/sentences. Modification is preferred to removing. But if a caption sentence, referring sentence, or question-answer pair is totally wrong/ambiguous/uncertain, remove it.

- For caption annotations: Make sure each piece of information in the caption is correct. Remove uncertain or meaningless elements. Be careful of object counts, take into account all objects, both referred and not referred.

- For object referring annotations: Make sure each referring sentence can distinguishably identify the correct object (numbered in boxes) without any ambiguity. Be careful of object color/orientation.

- For VQA annotations: Make sure each question has a clear answer without any ambiguity, and each answer should be correct using a single word or phrase, no more than 3 words. Correct answers that specify objects by object IDs. Remove self-answered question-answer pairs.

- Include question type for each QA, all possible question types include: object category, object existence, object quantity, object color, object shape, object size, object position,

object direction, scene type, and reasoning. The category of scene type includes color or grayscale, main structure/type of area, and rural or urban. Additionally, the category of reasoning is available for questions that require multifaceted analytical thought processes (e.g., object distribution pattern).

# 7 Experimental details

## 7.1 Training details

In our experimental setup, all comparative methods are trained on a single node equipped with 4 Nvidia A100 GPUs. The batch size is standardized at 32, and each model undergoes training for a duration of five epochs. We initialize the learning rate at 2e-4 and employ a cosine learning rate decay schedule for optimization. The learning rate experiences a warm-up phase, reaching 3% of the total training steps to gradually adapt to the training regime.

## 7.2 Visual grounding using OBBs

**Settings**. In the main paper, horizontal bounding boxes are utilized for both training the model and evaluating its visual grounding capabilities. This section extends the evaluation to incorporate oriented bounding boxes for object localization. Given that GeoChat has demonstrated superior performance in object grounding using bounding boxes, this experiment is exclusively dedicated to exploring the effectiveness of GeoChat under the conditions of oriented bounding boxes. In our experiments, the oriented bounding box is defined by the parameters $[cx, cy, w, h, \theta]$, where $(cx, cy)$ represents the center coordinates, $w$ and $h$ represent the width and height of the bounding box, respectively, and $\theta$ indicates the rotation angle.

**Results**. As shown in Table 2, the GeoChat model achieves an overall grounding accuracy of 24.3% at a threshold of 0.5, which is lower than its counterpart using horizontal bounding boxes, where the grounding accuracy reaches 49.8%. This highlights that visual grounding with oriented bounding boxes presents a greater challenge compared to grounding with horizontal bounding boxes.

Table 2: Visual grounding performance on VRSBench dataset using orientated bounding boxes for referring object localization. Err_R denotes average angle prediction error.

| Method | Unique | | Non Unique | | All | | |
|---|---|---|---|---|---|---|---|
| | Acc@0.5 | Acc@0.7 | Acc@0.5 | Acc@0.7 | Acc@0.5 | Acc@0.7 | Err_R |
| GeoChat [4] | 32.3 | 12.6 | 18.5 | 5.7 | 24.3 | 8.6 | 18.7 |

## 7.3 Qualitative results

We show qualitative results of detailed image caption in Fig. 1, visual grounding in Fig. 2, and visual question answering in grounding in Fig. 3.

## 7.4 GPT-4V Evaluation Prompt

"You are an AI visual assistant tasked with analyzing remote sensing images. You receive an image and multiple object-referring sentences and visual questions. Your role is to provide a detailed caption for the image, identify object locations for all referring sentences, and answer all visual questions.

Here are detailed instructions for image caption: Describe the image in detail in 3-7 sentences, making sure to include counts of prominent objects. Describe only clear features; avoid uncertainties and unclear elements. Do not mention anything that is unknown or not specified. Highlight diverse object attributes such as color, shape, position, size, relative position, and relative size. Do not add size details for small or large vehicles. Exclude imagined details not visible in the image, like weather, people, or object usage. Do not imagine the moving status of airplanes, ships, or vehicles if you are not sure about it. For roads, include features like shape (straight/curved), width, length, and orientation. For houses, mention characteristics like density, size, rooftop color, and presence of gardens. For airports, include details like boarding bridges, terminals, boarding ports, and tarmac.

(a)

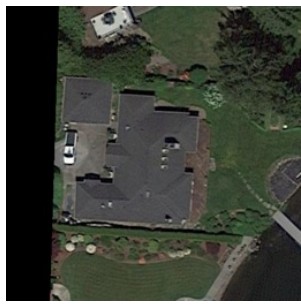

**GT**: The high-resolution image from GoogleEarth shows a large house with a complex roof structure, surrounded by a well-manicured lawn with landscaping features. There is a large vehicle parked on the left side.

**MiniGPT-v2**: 1080p high-resolution aerial image from googleearth showing a residential area with a large house surrounded by lush greenery and a small vehicle parked near the house the house has a distinctive dark roof and a driveway leading to the.

**LLaVA**: The image features a large house with a dark roof, surrounded by a well-maintained yard. There are two vehicles parked near the house: one is located towards the left side of the image, while the other is positioned more towards the center. The yard is adorned with a variety of potted plants, with some placed closer to the house and others scattered around the area.

**GeoChat**: The high-resolution image from GoogleEarth shows a residential area with a large house featuring a dark roof. The house is surrounded by a well-maintained lawn and is adjacent to a body of water. There is a small vehicle parked on the driveway near the house.

(b)

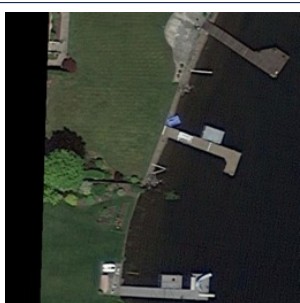

**GT**: The high-resolution image from Google Earth features a body of water with three small harbors. One harbor is located near the top right corner. Another is situated along the bottom edge. The third harbor is positioned between the other two, encompassing boat slips and a pier.

**MiniGPT-v2**: 2020 high-resolution image from googleearth showing a waterfront area with multiple harbors and a ship there are three harbors visible, with the largest one located at the top-right corner of the image the ship is positioned towards the.

**LLaVA**: The image features a high-resolution aerial view of a harbor with a large body of water. There are two harbors visible in the scene, one located towards the right side and another towards the left side of the image. The harbor on the right is larger and extends from the middle to the right edge of the image, while the harbor on the left is smaller and situated more towards the left side. In addition to the harbors, there is a small ship situated in the middle-left part of the image, close to the water's edge. The presence of the ship and the harbors suggests a bustling waterfront area.

**GeoChat**: The high-resolution image from GoogleEarth shows a waterfront scene with several harbors and a ship. There are three harbors visible, with the largest one located at the bottom of the image, extending from the center to the right edge. The smallest harbor is situated at the top-right corner, and the second harbor is positioned between the two, closer to the center. A unique ship is docked at the bottom-left harbor.

Figure 1: Selected examples of detailed image caption results. We highlight correct information in green and incorrect information in red.

Do not mention the image is taken during the day or night. Do not mention whether the vehicles are in motion or not. Carefully verify each piece of information before finalizing the caption.

For each referring sentence, tell me the location of the referred object in the image, and return its bounding box coordinates in the format of [x1, y1, x2, y2], which denotes the top left x, top left y, bottom right x, and bottom right y.

Here are detailed instructions for visual question answering: For each question, answer the question based on the image content in a single word or a short phrase.

Finally, you need to return {caption: detailed image caption, objects: [obj_id, referring_sentence, location,...], qa_pairs: [ques_id: question id, type: question type, question: question, answer: answer]} in JSON format. Do not return any notes after the JSON."

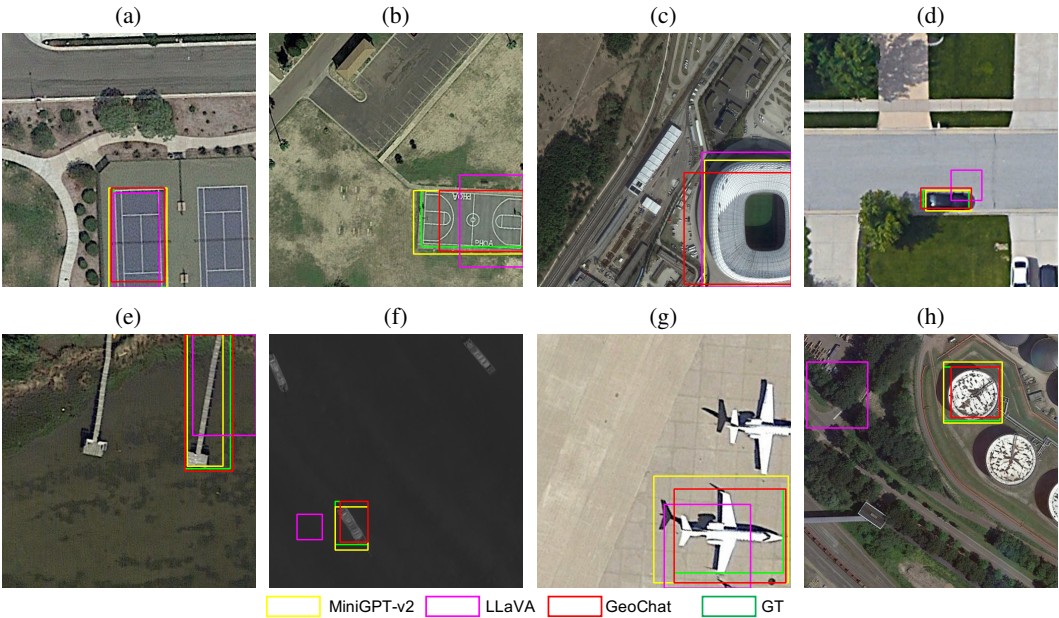

Figure 2: Selected examples of Visual grounding. (a) The tennis court on the left side of the image, surrounded by a brownish surface. (b) The basketball court located at the right side of the image. (c) The dome stadium situated towards the bottom-right side of the image. (d) The vehicle parked closest to the top edge of the image. (e) The harbor located at the right-most edge of the image. (f) The small ship located towards the bottom-left of the image. (g) The airplane is located towards the bottom of the frame. (h) The left-most storage tank is fully visible and situated on the upper side of the image.

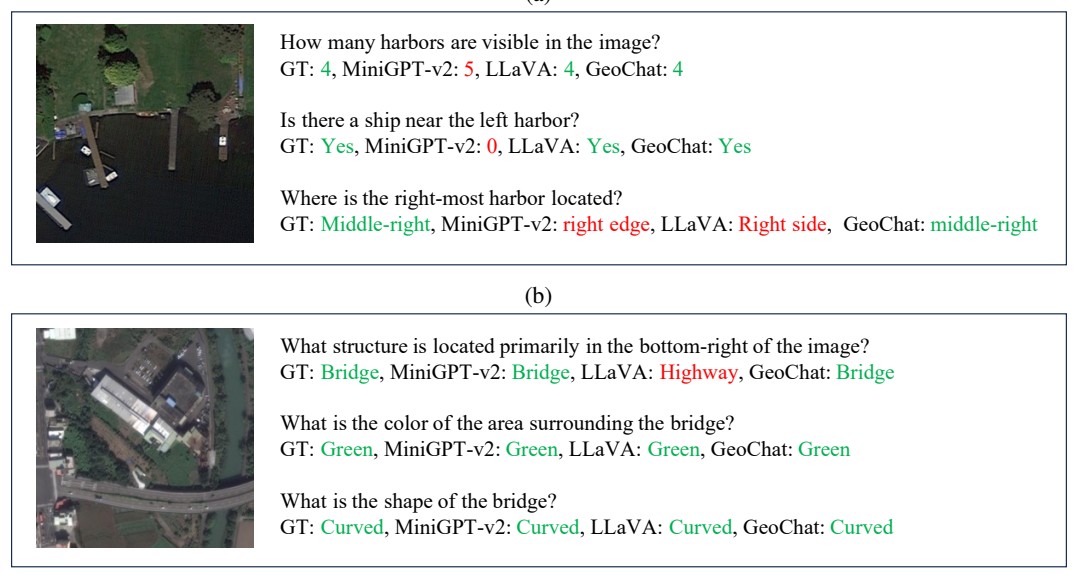

Figure 3: Selected examples of VQA results. Correct answers are shown in green and incorrect answers are shown in red.