# OpenReview forum: "VRSBench: A Versatile Vision-Language Benchmark Dataset for Remote Sensing Image Understanding"
_NeurIPS.cc/2024/Datasets_and_Benchmarks_Track — NeurIPS 2024 Track Datasets and Benchmarks Poster_

### Official Review · Reviewer_YF6C · 2024-07-24
**A High-Quality Remote Sensing Dataset**

**Rating:** 7
**Confidence:** 4

**Review:**

The paper introduces VRSBench, a high-quality dataset for remote sensing with thorough documentation and ethical data practices. It supports multiple vision-language tasks and provides detailed, human-verified annotations. The work is clear, original, and significant. However, it's limited to RGB images and the human verification process is costly. Evaluating more models could provide broader insights.

**Strengths:**

1. VRSBench provides detailed, human-verified annotations, ensuring high quality of remote sensing dataset.
2. The dataset supports a variety of tasks, facilitating the development of versatile models.
3. This paper evaluates existing models on VRSBench, providing a valuable benchmark.

**Additional Feedback:**

None.

**Clarity:**

The paper is well-written and clearly structured. The methodology, dataset, and evaluation procedures are explained in detail.

**Correctness:**

The paper is well-researched and methodologically sound. The semi-automatic pipeline and human verification ensure data quality. Evaluation metrics and methods are standard and appropriate.

**Documentation:**

The documentation is clear and thorough, with detailed explanations of the methodology and dataset creation. The inclusion of prompts, guidelines, and code ensures replicability. More details on hyperparameters and hardware used would be helpful.

**Ethics:**

Ethical considerations are well-addressed. The paper discusses societal impacts and ensures no personally identifiable information is included. The human verification process is ethical, with proper compensation. The dataset and code are openly available under a permissive license.

**Limitations:**

same as Opportunities For Improvement

**Opportunities For Improvement:**

1. The authors seem to evaluate similar MLLM models (model size, training data). Hopefully the authors can include a wider range of models in the evaluation to provide a richer evaluation criterion.
2. The current dataset only contains RGB images, limiting its applicability to other remote sensing data such as infrared and SAR images.
3. A deeper analysis of the model performance on specific tasks would be helpful.

**Relation To Prior Work:**

The paper compares VRSBench with existing remote sensing vision-language datasets, highlighting its advantages in data quality and task diversity. Evaluating state-of-the-art models on VRSBench establishes its relevance and impact.

**Summary And Contributions:**

The paper introduces a new benchmark dataset aims to advance vision-language models for remote sensing images. The dataset contains about 30K remote sensing images and has been manually verified. It is worth noting that the dataset supports various  task types of MLLM, such as image captioning, visual grounding, and visual question answering, and evaluates the state-of-the-art models of these tasks.

---

> ### Author Rebuttal · Authors · 2024-08-16
>
> Q1: Evaluation only on similar MLLM models (model size, training data).
>
> A: We evaluate MLLM models trained on the same data and of comparable size to ensure a fair comparison. This approach is essential because the vision-language capabilities of MLLMs are closely linked to their model size. For instance, LLaVA-1.5-7B and LLaVA-1.5-13B achieve quite different VQA accuracies of 58.2% and 61.3%, respectively, on the TextVQA dataset.
>
> Q2: The current dataset only contains RGB images.
>
> A: We acknowledge this limitation in our dataset and plan to enhance VRSBench by incorporating annotations from a broader range of remote sensing data types, including infrared, multi- and hyperspectral images, Synthetic Aperture Radar (SAR) images, and temporal datasets. This is further discussed in Section 6, 'Conclusion and Future Work.'"
>
> Q3: Deeper analysis of the model performance on specific tasks.
>
> A: Thank you for your suggestion. However, due to page limitations, we are unable to provide a detailed discussion of the experimental results in this submission. If the paper is accepted, we will include more detailed discussions in the camera-ready version.

---

> > ### Author Response · Authors · 2024-08-28
> >
> > Dear Reviewer,
> >
> > Thank you once again for your valuable comments. As the discussion period nears its deadline, we are more than happy to address any last-minute questions or clarify any remaining doubts regarding our paper. If our responses have adequately addressed your concerns, we kindly request that you consider increasing the scores.
> >
> > Thank you and best regards,
> > The Authors

---

### Official Review · Reviewer_YFnT · 2024-07-24
**Review for VRSBench**

**Rating:** 7
**Confidence:** 4
**Clarity:** The paper is well-organized and easy …

**Review:**

Pros:
* Presents a useful benchmark for future LVLM evaluation.
* The dataset is well documented.

Cons:
* The automatic steps of the pipeline need to be studied more.
* GPT-4V and other proprietary models are not tested.

Overall the paper provides a useful and novel dataset and therefore my recommendation is an accept. However,  the authors should address the following concerns.

**Strengths:**

The paper provides an important dataset for remote sensing LVLMs which the a direction the community is heading towards. Therefore the dataset is timely.

Significant human annotation effort has been put into the dataset to make the contributions important.

The dataset is easy to access (already on hugging face), well-documented and therefore should be easy to use for future work.

**Additional Feedback:**

Line 97: mothods: methods

In table 1 caption and figure 3, the term "word length" is a bit confusing. "word length" signifies the number of characters in a word which is not what is being quantified. Use "Word count" or "Number of words" instead.

In the last column of table 1 it is unclear what Human means. Does it mean how much annotation work is done by humans vs how much work is automatic? If so, shouldn't VRSBench have a yellow tick as a big part of it is done automatically? The meaning of the column should be specified clearly in the caption.

**Correctness:**

The dataset is constructed in a sound way. One small concern is that since the dataset is generated via an LVLM, The biases in the text need to be studied. Would the captions generalize to non-western satellite image concepts.

**Documentation:**

The dataset is well documented.

**Ethics:**

The are no major ethical concerns.

**Limitations:**

The limitations of the paper are **not** discussed adequately. The authors talk about expanding the image domain b adding other bands and sources as future work, however this seems insufficient.

**Opportunities For Improvement:**

It is unclear from reading the paper how much correction is needed from humans when inspecting the images. While GPT-4v is relatively good at understanding satellite images, it should be clarified how many times do humans make corrections and what kind of corrections do they make.

In a similar vein, it might be useful to know the performance of GPT-4V on this dataset (or at least a subset). This is to make sure that GPT-4V is using the visual information or information from annotations to create the dataset.

Since a big step of the dataset is automatic, it might be useful to understand the biases generated due to the automatic. See correctness for biases of the language model. But also, does a model trained on a subset of objects in VRSBench training set generalize to unseen objects or does it overfit to the objects in the training set.

**Relation To Prior Work:**

Prior datasets and the related methods for all the tasks are well discussed.

**Summary And Contributions:**

The paper presents a dataset for Remote sensing LVLM evaluation. It presents data for three tasks image captioning, visual grounding, and VQA. The dataset is built on images and annotations from previous object detection datasets such as DIOR. It uses GPT-4V to provide detailed captions, QA pairs, and object references. In total, there are about 50k images which is significantly larger than the prior dataset of this type. The key contribution of the paper is it uses object annotations and images as input to GPT-4v to create the dataset and uses human verification to validate. For evaluation, the paper fine-tunes and tests three open-source LVLM models on this dataset.

---

> ### Author Rebuttal · Authors · 2024-08-16
>
> Q1: Numbers and what kind of corrections do human annotators make?
>
> A: The reviewer raises a good point. Although GPT-4V can generate annotations at descent quality, it still tends to make certain mistakes. On average, the human annotators made 2~4 corrections for GPT-4V-generated annotations per image. Common corrections made by human annotators include:
> 1) incorrect object counts when many objects appear;
> 2) incorrect object colors, since this information is not provided to GPT-4V;
> 3) uncertain information;
> 4) hallucinated information, such as weather, people, or object usage;
> 5) ordinal descriptors and references.
>
> Q2: The performance of original GPT-4V.
>
> A: We have already provided the results of the original GPT-4V in the supplementary material. Please refer to Tables 2-4 in the supplementary. From these results, GPT-4V demonstrates decent performance in detailed image captioning and visual question answering, indicating that GPT-4V can understand satellite images to a certain degree. However, due to the absence of object information, GPT-4V performs poorly in the visual grounding task.
>
> Q3: Biases in the text and generalization to non-western satellite image concepts.
>
> A: We acknowledge that GPT-4V-generated annotations can have certain biases towards geographic locations. Such biases depend on the datasets used to train GPT-4V. Since the training dataset and model of GPT-4V are close-set, it's hard to evaluate the bias of GPT-4V without human study. Moreover, since we use detailed object information as a prompt to GPT-4V, we actually do not observe serious biases in GPT-4V-generated annotations.
>
> Q4: Generalization to unseen object categories.
>
> A: Since our annotations contain information beyond original DOTA and DIOR annotations, such as roads, houses, and trees, the model trained on our dataset can therefore generalize to these "unseen" object categories. For object categories not included in our VRSBench annotations, since the vision backbone in LLaVA, e.g., CLIP, has open-world understanding capabilities, we believe the finetuned still has generalization capabilities to unseen object categories.
>
> Q5: grammar issues
>
> 1) Line 97: mothods: methods.
>
> A: Thanks for your suggestions, we will correct all grammar issues in our revised version.
>
> 2) In Table 1 caption and Figure 3, the term "word length" is a bit confusing. "word length" signifies the number of characters in a word which is not what is being quantified. Use "Word count" or "Number of words" instead.
>
> A: Thanks for your suggestions, we will polish the language in our revised version.
>
> 3) What Human means in Table 1.
>
> A: The column "Human" indicates whether the collected annotations for each dataset are verified by human annotators or not. The RSIVQA dataset is marked with a yellow tick because only a small portion of question-answer pairs in RSIVQA are annotated by human annotators. In comparison, even though part of our dataset collection process is done automatically, all annotations in our VRSBench dataset are verified by human annotators.

---

> > ### Author Response · Authors · 2024-08-28
> >
> > Dear Reviewer,
> >
> > Thank you once again for your valuable comments. As the discussion period nears its deadline, we are more than happy to address any last-minute questions or clarify any remaining doubts regarding our paper. If our responses have adequately addressed your concerns, we kindly request that you consider increasing the scores.
> >
> > Thank you and best regards,
> > The Authors

---

### Official Review · Reviewer_5s1x · 2024-07-25
**New vision-language remote sensing dataset.**

**Rating:** 7
**Confidence:** 4

**Review:**

See the strengths and weaknesses below.

**Strengths:**

1. The paper provides a vision-language dataset for multiple applications in the field of remote sensing, developing three benchmarks, including image captioning, visual grounding and visual question answering.

2. The dataset has been verified by human annotators, which ensures its utility for real applications.

3. Comparison between the proposed VRSBench dataset and the existing remote sensing vision-language datasets shows the advantages of VRSBench. Also, the pipeline of constructing VRSBench and the details of data creation are clearly introduced.

4. This paper presents a benchmark evaluation of cutting-edge models, demonstrating the dataset's utility as evidenced by the models' enhanced performance following fine-tuning on VRSBench.

**Additional Feedback:**

No additional feedback at this point.

**Clarity:**

Besides the issue in the "Opportunities for Improvement", the paper is well written.

**Correctness:**

Besides the limitations above, the datasets and the benchmark are constructed in a sound way.

**Documentation:**

Yes, there are links to the github repo and datasets.

**Ethics:**

No ethical concerns.

**Limitations:**

1. From L74 and Table 1, could the authors specify the definition of “open-set” in this study? I wonder if it means the questions can include the objects not provided in the original categories; however, these categories can be both included in the training set and test set. Clarification on this matter would be beneficial.

2. Table 1: Since the label of DIOR dataset is horizontal bounding box, do all the 52,472 references in the VRSBench correspond to oriented bounding boxes? From Section 7.3, it is not clear whether the training set of VRSBench also evolves OBB. This issue should be stated more clearly.

**Opportunities For Improvement:**

1. Given the diverse range of annotations provided by VRSBench, it is recommended to test the model fine-tuned on VRSBench on other existing datasets, as many are listed in Table 1.

2. L196: What are the loss functions for the three tasks, and how can the optimization be balanced among them, given the varying numbers of captions, visual references, and question-answer pairs?

3. L255: “Further finetuning on our VRSBench training set significantly boost…”, “boost” should be “boosts”. L206:” benif of finetuning”, please check the spelling.

4. L224 in the supplementary material “Nvidia 100 GPUs”, please specify the GPU model (V100) for clarity.

**Relation To Prior Work:**

The literature review is comprehensive. Also, the evaluation of the recent related methods, such as GeoChat, has been conducted using the dataset.

**Summary And Contributions:**

This paper introduces a new vision-language benchmark dataset for image caption, visual grounding and visual question answering on remote sensing imagery. The contribution is clear by addressing the problems of the existing vision-language dataset in remote sensing domain.

---

> ### Author Rebuttal · Authors · 2024-08-16
>
> Thank you for providing the valuable feedback. Please find our answers to your questions below.
>
> Q1: Performance of fine-tuned models on other existing datasets.
>
> A: This is an interesting point. According to your suggestion, we evaluate the performance of the fine-tuned model on a visual grounding dataset (i.e., DIOR-RSVG) and several visual question-answering datasets (e.g., RSIVQA). We report the performance below. From the tables, the GeoChat model trained on VRSBench demonstrates reasonable zero-shot performance on visual grounding and VQA tasks. We will include these results in our final version.
>
> Table 1. Visual grounding performance on DIOR-RSVG dataset. The top part shows supervised training performance, and the bottom part shows zero-shot generalization performance. * indicates the model trained on VRSBench.
>
> | Method     | Acc@0.5 | Acc@0.7 |
> |------------|---------|---------|
> | ZSGNet     | 51.7    | 42.3    |
> | TransVG    | 72.4    | 60.1    |
> | MGVLF      | 76.8    | 66.7    |
> | EarthGPT   | 76.7    | 66.5    |
> | GeoChat*   | 42.5    | 19.1    |
>
> Table 2. Visual question answering performance on CRSVQA, RSVQA, and RSIVQA datasets. The top part shows supervised training performance, and the bottom part shows zero-shot generalization performance. * indicates the model trained on VRSBench. Numbers in brackets indicate the performance evaluated based on GPT-4.
>
> | Method         | Open-ended | CRSVQA         | RSVQA-HR       | RSIVQA         |
> |----------------|----------|----------------|----------------|----------------|
> | RSVQA          | No       | 59.0           | 74.2           | 71.6           |
> | RSVQA (GRU)    | No       | 59.4           | 76.7           | 72.2           |
> | MQVQA          | No       | 70.9           | 82.2           | 75.5           |
> | GeoChat        | Yes      | -              | 72.3           | -              |
> | EarthGPT       | Yes      | 82.0           | 72.1           | -              |
> | GeoChat*       | Yes      | 27.7 (57.7)    | 50.4 (50.4)    | 50.1 (72.4)    |
>
>
> Q2: Loss functions and task balance.
>
> A: For model training, we employ token-wise cross-entropy loss across all three tasks. In our previous experiments, we did not attempt to balance these tasks; instead, we jointly trained on all tasks with equal weighting. As our primary focus is on constructing a benchmark dataset rather than model design, we have left this aspect for future exploration.
>
> Q3: writing issues:
> 1) “boost” should be “boosts”.
> 2) L206:” benif of finetuning”, please check the spelling.
> 3) specify the GPU model (V100) for clarity.
>
> A: Thank you for your attention to detail. We will address all grammar and writing issues in the revised version. We will include the specific GPU model (A100) for clarity in the revised version.
>
> Q4: Clarification on open-set in Table 1.
>
> A: Apologies for the confusion. We actually meant "open-ended" rather than "open-set." Open-ended refers to question-answer pairs presented in free-form text, as opposed to being selected from predefined question/answer sets (i.e., closed-set). In contrast, RSVQA contains closed-ended (multiple-choice) question-answer pairs, with examples of answers such as [yes, no, 1, 2, 3...].
>
> You are correct in noting that "questions can include objects not provided in the original categories." Our annotations extend beyond the original DOTA and DIOR annotations to include additional information such as roads, houses, and trees. This is one of the key strengths of our dataset, and we will include a clarification in the revised version. Thank you for bringing this to our attention.
>
> Q5: Regarding oriented bounding boxes in the DIOR dataset.
>
> A: When collecting annotations for the VRSBench dataset using GPT-4V with human verification, we utilized the DIOR dataset with Horizontal Bounding Boxes (HBB). The experiments presented in the main paper were conducted using the HBBs from this dataset.
>
> After submitting the main paper, we discovered that the enhanced version of the DIOR dataset, known as DIOR-R, also includes Oriented Bounding Boxes (OBBs). Consequently, we developed a new version of our dataset by replacing HBBs with OBBs for all objects in the DIOR dataset. This was achieved by matching objects in DIOR and DIOR-R based on their locations. We then reran the visual grounding experiments using OBBs, and the results are reported in Section 7.3 of the supplementary material.
>
> This is also documented in our Huggingface repository before the supplementary submission deadline:
> "Note that the original DIOR dataset uses HBB to localize objects, we further convert HBBs to OBBs in our annotations (based on DIOR_R dataset), please check the dior_r branch for annotations".

---

> > ### Comment · Reviewer_5s1x · 2024-08-31
> > **Post-rebuttal discussion**
> >
> > The author has addressed my concerns (and carried out the fine-tuning experiment on other datasets). Therefore, I am considering maintaining the rating.

---

> ### Author Response · Authors · 2024-08-28
>
> Dear Reviewer,
>
> Thank you once again for your valuable comments. As the discussion period nears its deadline, we are more than happy to address any last-minute questions or clarify any remaining doubts regarding our paper. If our responses have adequately addressed your concerns, we kindly request that you consider increasing the scores.
>
> Thank you and best regards,
> The Authors

---

### Author Rebuttal · Authors · 2024-08-16

Firstly, we would like to express our sincere gratitude to the reviewers for taking the time to evaluate our paper and for providing constructive feedback. We are encouraged by the acknowledgment from all reviewers of the novelty and contribution of our newly introduced VRSBench dataset. Specifically, Reviewer YF6C finds our work to be clear, original, and significant. Reviewer YFnT appreciates that our work presents a useful and novel benchmark for future LVLM evaluation, and Reviewer 5s1x states that the contribution is clear.

The primary concerns raised relate to annotation biases introduced by GPT-4V, implementation details (such as the loss function and task balance), clarifications on OBB and open-set properties, as well as grammar and writing issues. We address all of these concerns in the following sections.

Additionally, we identified a bug during the preparation of the training set for three baseline models—LLaVA-1.5, Mini-Gemini, and GeoChat. This issue caused only one object and one question-answer pair to be selected for training, while others were ignored. As a result, the performance of these models trained on the VRSBench set was affected. We have rerun all experiments with the corrected dataset and reported the updated performance in the rebuttul.pdf. The visual grounding and question answering performance of fine-tuned models improved by 5-10% with the new training set. We will update these numbers in our revised version.

---

### Decision · Program_Chairs · 2024-09-26

**Decision:**

Accept (Poster)

**Comment:**

This paper makes a solid contribution to VLM-based remote sensing tasks. The proposed dataset is very promising and helpful to the development of this subfield. All reviewers agree with the major contributions of this paper.